# Changes in Stereotypies: Effects over Time and over Generations

**DOI:** 10.3390/ani12192504

**Published:** 2022-09-20

**Authors:** Patricia Tatemoto, Donald M. Broom, Adroaldo J. Zanella

**Affiliations:** 1Center for Comparative Studies in Sustainability, Health and Welfare Department of Veterinary Medicine and Animal Health, School of Veterinary Medicine and Animal Science, FMVZ, University of São Paulo—USP, Pirassununga 13635-900, SP, Brazil; 2Department of Veterinary Medicine, Centre for Animal Welfare and Anthrozoology, University of Cambridge, Cambridge CB3 0ES, UK

**Keywords:** welfare, emotionality, gestation, brain change, stereotypy, stereotypic behavior

## Abstract

**Simple Summary:**

Herein, we propose that there should be discussion about the function and effects of stereotypies in relation to the time during which they are shown. In the first stages, stereotypies may help animals deal with challenges. However, behavior can potentially alter the brain, impairing its function due the absence of a diverse repertory, and change brain connections, neurophysiology and later neuroanatomy. The neuroanatomical changes in individuals showing stereotypies could be an effect rather than a cause of the stereotypy. As a consequence, studies showing different outcomes for animal welfare from stereotypy expression could be due to variation in a timeline of expression. Stereotypies are widely used as an animal welfare indicator, and their expression can tell us about psychological states. However, there are questions about the longer-term consequences if animals express stereotypies: do the stereotypies help in coping? During the prenatal period, stereotypic behavior expressed by the mother can change the phenotype of the offspring, especially regarding emotionality, one mechanism acting via methylation in the limbic system in the brain. Are individuals that show stereotypies for shorter or longer periods all better adjusted, and hence have better welfare, or is the later welfare of some worse than that of individuals that do not show the behavior?

**Abstract:**

Stereotypies comprise a wide range of repeated and apparently functionless behaviors that develop in individuals whose neural condition or environment results in poor welfare. While stereotypies are an indicator of poor welfare at the time of occurrence, they may have various consequences. Environmental enrichment modifies causal factors and reduces the occurrence of stereotypies, providing evidence that stereotypies are an indicator of poor welfare. However, stereotypy occurrence and consequences change over time. Furthermore, there are complex direct and epigenetic effects when mother mammals that are kept in negative conditions do or do not show stereotypies. It is proposed that, when trying to deal with challenging situations, stereotypies might initially help animals to cope. After further time in the conditions, the performance of the stereotypy may impair brain function and change brain connections, neurophysiology and eventually neuroanatomy. It is possible that reported neuroanatomical changes are an effect of the stereotypy rather than a cause.

## 1. Introduction

Stereotypies constitute a range of repeated and apparently functionless behaviors that are expressed by individuals whose neural condition or environment results in compromised welfare [1,2,3] and occur in situations wherein an individual lacks control of their environment [4,5]. Many factors can be involved in the causation and development of stereotypies. Moreover, the behavior is apparently functionless at the time of occurrence but could have positive or negative consequences. Stereotypies were described as a invariant sequence of movements occurring frequently, in a specific context, which could not be considered part of the functional systems of the animal [1]. Furthermore, it is described as a behavior that develops in an environment that causes poor welfare, is repetitive, does not vary and apparently has no function. Stereotypies may also occur in individuals with neurological disorders, and one of the definitions is a repeated, relatively invariant sequence of movements that has no obvious purpose [5,6]. Moreover, studies of the causation of frequent stereotypies in sows (*Sus scrofa domesticus*) kept in stalls, crates or tethers [1,7,8] suggest that bar-biting may start as an escape attempt, drinker-pressing as an effort to control when food can be obtained and sham-chewing as a replacement for actual eating [6]. However, it is likely that these behaviors soon become functionally divorced from these aims. At this time, they still indicate that the environment is very negative and that welfare is poor. The word function is not used in the definition, because an immediate function of the movement may be apparent, while the long-term purpose is not.

Environmental enrichment can be an efficient strategy to reduce the expression of stereotypies [9,10,11], so the changed experience must have an effect on the causal factors associated with the continuation of these behaviors. Enrichment implies an improvement that meets the needs of the captive animal species better than without it, so there are conditions for better biological functioning [9]. If the needs of the animals are met, they will not show indicators of poor welfare, such as stereotypies. Additionally, the possibility of interacting with more complex environments is the context in which the motivational systems have been selected for from an evolutionary perspective.

One of the hypotheses for the causation of stereotypies assumes that their performance helps the individual to cope with its environment and reduces distress in the animal [12,13]. It is clear that the stereotypy is an attempt to cope, but does it actually help in coping? While it seems likely that some stereotypies help, some can be damaging to the individual. Headshaking in domestic fowl (*Gallus gallus*) is an example of a stereotypy that probably helps to regulate attentional mechanisms [5] but is an indicator of poor welfare if repeated too frequently. Several normal and abnormal behaviors can be used to affect motivational state. The fact that an abnormal behavior is adaptive in the short-term does not mean that it is not an indicator of poor welfare at the time of occurrence [6].

A stereotypy in a mother animal could be either positive or negative in relation to attempts to cope but could also affect the offspring. The prenatal environment can change the characteristics of a fetus in ways that persist after birth. In mammals, pregnancy has a potential role in shaping the ontogenetic development of the organism, once the environment of the mother may have consequences on the offspring [14,15,16].

Neurodevelopmental programming can induce alterations that will impact how the animal will cope with challenges in the postnatal environment [17]. Epigenetic mechanisms modulate the phenotype by fetal programming, in which the modifications are stable and cross into further generations. In this context, the effects in the offspring may depend on the causes and time course of stereotypies in the mother. It could be that the welfare of the offspring from mothers that show the behavior have different welfare from the ones that do not show stereotypies. The role of experience during the time-course of stereotypies on individuals and their offspring is reviewed here.

## 2. Stereotypies as Animal Welfare Indicators

Stereotypies develop in animals kept in environments where they have little control, sometimes with few stimuli, physical restraint or exposure to contingencies that cause fear or frustration [3]. Stereotypies have been described in a wide range of species maintained in artificial environments. The occurrence of stereotypies suggest frustration related to the inability to perform highly motivated behaviors, which may be tentatively expressed even in the absence of appropriate stimuli. However, the expression of stereotypies is a consequence of an interaction between the control systems of the individual and the environment that it encounters.

The expression of stereotypies is considered to be an indicator of poor welfare [2,3,4,5,18,19]. Since stereotypies occur when individuals have long-term problems, there is usually no associated increase in levels of cortisol [6,20,21]. Most scientists would say that there is no reason to suppose that stereotypies will affect physiological systems associated with stress response [13]. Situations inducing or exacerbating stereotypies lead to poor welfare [22] and are often associated with frustration [2,23]. Stereotypies often occur in barren environments, and the reduced level of stimulation in these conditions leads to boredom-like states [24]. The high predictability of the environment, typical of barren environments, can be a causal factor triggering stereotypies [25].

Among the neurophysiological markers implicated in these behaviors are endorphins, and the opioid antagonist naloxone interrupts stereotypies in sows [26,27]. Endorphins may influence stereotypies that are developing, since stereotypies that have continued for a long time are not sensitive to naloxone treatment [26]. Moreover, endogenous opioids may be involved in the positive feedback and could be the reason for the maintenance of the behavior and inhibition for switching to others, decreasing behavioral flexibility [27,28]. Mice (*Mus musculus*) in barren housing conditions showed an alternative to stereotypic behavior, a motionless behavior considered to be a depression-like state [29], similar to that reported for confined sows and horses [29,30]. These data do not conflict with the idea that the performance of stereotypies can help animals deal with challenging contexts [12,22,27]. However, the responses to the condition, and their consequences, are likely to change over time (see Figure 1). In the first stages, stereotypies may help animals cope with challenges, but later they may not. It is likely that the performance of the behavior can alter the brain, impairing its function due to the absence of a diverse repertory, and change brain connections, neurophysiology and eventually neuroanatomy. Some changes in the brain were described in previous studies [8,31], but it is not clear if it is the cause or consequences of stereotypy expression.

The frustration triggered by food restriction can initiate oral stereotypic behaviors in sows [32]. The expression of stereotypies has also been investigated in relation to some factors, such as the genetic component [33,34], personality predisposition [35,36], individual variation [37] and susceptibility in relation to sex [38]. Some environmental variables have more impact on the occurrence of stereotypies than others [39]. The motor patterns of wheel-running in mice fit the concept of stereotypy, and this may help to explain the causation of other behaviors [37]. Regarding cognitive bias, mice with higher levels of stereotypic behavior made more optimistic choices [40].

Stereotypies were considered to be a pathological outcome related to the dysfunctional activity of the dopaminergic system, resulting from an overproduction of dopamine or a hypersensitivity in the receptors of this neurotransmitter [41]. Stereotypies may represent the appetitive phase of the motivational systems, in which the restrictive and barren environment does not permit the individual to reach the consummatory part of the behavior. The impossibility to reach the last phase of the motivational system may generate the repetition that characterizes stereotypies.

Interestingly, voles (*Clethrionomys glareolus*) showing stereotypies change their preferences, manifesting a bias for poor environments [42]. Correspondingly, wild voles kept in laboratory cages presented highly repetitive locomotor stereotypies and showed a preference for a less-enriched condition when compared with non-stereotyping voles [42]. One possible way to elucidate these findings may be the fact that it would be physically easier to perform stereotypies in a barren environment [42].

Information about stereotypies comes mainly from animals in captivity and from individuals with neurological disorders. What are the consequences of stereotypies in the long-term? Male mink (*Neovison vison*) that showed a longer duration of stereotypies had lower success in copulation [43]. This comparison was of animals that developed stereotypies in environmentally enriched conditions [43]. In this context, the environment could have impacted the ontogeny, adrenal functioning, reproductive mechanisms regarding physiology, social behavior or flexibility.

Opioid antagonists such as naloxone may reduce the frequency of stereotypies. These findings prompt suggestions that stereotypies increase endogenous opioid activity and thus induce a degree of analgesia in animals [26]. Nevertheless, this assumption was questioned [2,23] and was not supported by a study investigating this mechanism in horses [44]. Moreover, stereotypies in mink are associated with increased hippocampal neurogenesis [45]. It has been know that, under chronic stressful conditions, neurogenesis is decreased [46,47]. Taken together, these outcomes could indicate a relation with less stress response in animals expressing stereotypies. In horses, stereotypies can be associated with the consequences of an ACTH challenge test, but it is not clear how this relates to coping strategies [48]. Mink that chew their own tails may also explore more [49], but this does not tell us that repetitive tail-chewing is good for the mink. Similarly, increased reproductive output in female mink performing stereotypic behavior [34,50] is not simple to interpret. Positive correlations between fertility, incidence of stereotypies and bodyweight in mink [34] tell us something about the survival strategies of mink in small cages but do not tell us that welfare is good in these cages. The mink that shows abnormalities other than stereotypy may reproduce less well, but stereotypy always indicates that the individual has a serious problem. The welfare of confined sows that show apathy and unresponsiveness may well be worse than those that show stereotypies, but neither has good welfare.

Some authors have suggested that stereotypy in animals maintained in artificial environments is related with some level of abnormality which leads to a brain dysfunction similar to that in schizophrenia and autism spectrum disorder [51]. Tail-biting in pigs, feather-pecking in poultry and tail-chewing in mink involve some repetition but are not stereotypies, and both the actions and the consequences are completely different from stereotypies.

## 3. Why Environmental Enrichment Reduces Stereotypies

A wide range of studies shows that environmental enrichment reduces the occurrence of stereotypies [52,53,54]. The reduction of stereotypy expression is considered the main criterion for interpreting the enrichment successful [29,53]. Correspondingly, environmental enrichment is considered effective to improve biological functioning and the welfare of the individual [9]. When an environmental change really is enrichment, it tackles the cause of the problem by providing the environmental conditions that meet the needs and hence reaches the motivation of the animals [54]. There are some individual variations about the benefits of each kind of enrichment, since some individuals will be more prone to have positive effects from social enrichment, while others need physical or sensory stimulation.

Environmental enrichment early in life can “protect” against the later development of stereotypies [55]. The benefit for the individual is not the reduction in the performance of stereotypies but reduction in the frustration that triggers the occurrence of stereotypies. In other words, it has to do with the cause of the stereotypies. Environmental enrichment, in addition to allowing the expression of preferred behaviors, reduces endocrine and behavioral reactivity to challenging situations [56]. It can also alleviate cognitive and behavioral impairments [57,58], modulate aggressiveness [59], alter HPA axis activity [56,60,61,62,63], increase brain plasticity [11,56,62,64], increase performance of hippocampal mediated tasks [65,66,67] and reduce methylation in hippocampal and frontal cortex genes [68].

It is reasonable to assume that chronic stress and environmental enrichment act on similar mechanisms in the hippocampus, so the enrichment could enhance brain activity and optimize resilience under chronic stress conditions [69]. For instance, the plasticity in hippocampal astrocytes when there is environmental enrichment is associated with the protection in the brain, mitigating cognitive impairments related to age, improving spatial memory and inducing accurate spatial strategy [70].

Taken together, it is expected that environmental enrichment affects the expression of genes in the brain, especially those involved in neuronal structure, synaptic signaling and plasticity [71]. Some of these genes described in previous studies are known to be associated with learning and memory [72]. Additionally, the positive impacts of environmental enrichment includes the effects on brain weight, increasing in arborization and the density of dendritic spines [73], modulates neurogenesis in the hippocampus [74] and can make cognitive bias positive [10,75]. It also reduces anxiety [11,65,76,77], and enriched animals are likely to be less emotionally reactive in novel situations, so they can explore their environment more efficiently [56].

Male mice housed with enrichment after weaning gained less weight, displayed increased social behavior and presented lower corticosterone concentrations and prefrontal IL-1β elevations in response to a mild social stressor. Additionally, they exhibited reduced TNF-α and increased brain-derived neurotrophic factor (BDNF) expression in the pre-frontal cortex [78]. In pigs, enrichment increased BDNF in blood [79], and BDNF has been linked with increased stress resilience [80], since enhanced cognitive functions is implicated in growth, maintenance and plasticity in the brain [81]. In this case, the stress resilience is related to a buffer effect promoted by the environmental enrichment when animals are challenged, which agrees with other studies [75,82,83].

The benefits of sensory stimulation as enrichment [84,85] raise the argument that it is necessary to stimulate the brain, rather than simply to promote preferred behaviors. It is reported that environmental enrichment induces several neuroanatomical, neurochemical and behavioral impacts. Stimuli provided by enriched environments alter brain functioning, since the brain requires triggers to make or lose connections. This is mandatory for the healthy functioning of the brain. Losing connections is deleterious, and mechanisms may have evolved to shape the brain to preserve its functional structure.

## 4. Are There Consequences in the Offspring?

Could there be transgenerational effects of stereotypies with changes in offspring phenotype? Pregnancy plays an important role in shaping the organism in mammals, since the maternal environment may affect offspring development. The “thrifty phenotype Hhypothesis” concerns the neurodevelopmental reprogramming that induces alterations in the fetus to help cope with early life and additionally anticipate the postnatal environment [17]. This means that the prenatal environment has the potential to modulate the offspring’s phenotype and to prepare individuals to cope with challenges. The maternal conditions during gestation may result in changes in multiple offspring factors [14,15,16,86,87,88,89,90,91,92]. By this mechanism, characteristics such as emotional reactivity, responsiveness to stressors and cognitive functioning can be shaped by challenges in both prenatal and neonatal periods [14,93,94]. There is evidence that stressors, for instance, negative interactions with the handler [14,15,88] and social challenges [14], can alter emotional reactivity, social behavior, responsiveness to stressors, cognition and memory in offspring. Since sows experience food restriction during gestation, adding fiber and promoting less hunger affects the offspring positively, reducing aggressive behavior [16].

Contingencies in the early life can shape their phenotype and promote changes in their biological functioning. Prenatal environments are critical for fetal development, including the organization of the central nervous system. When mothers are stressed, there can be a wide range of effects on offspring physiology and behavior, since the systemic physiological stress response of the mother has connection to the offspring that is in development in the uterus. The environment of the mother can provide important signals to the fetus, enabling some later adjustments to the environment that will be encountered [68,94,95,96]. Confined captive animals may give harmful cues to the fetus due to the inadequacies of the environment that imposes a dysfunctional and extreme lifestyle, in which for instance, commonly, the environment makes it impossible for the animal to perform very basic activities, such as to locomote [14,15,97].

On the other hand, prenatal stress may generate changes that are not necessarily pathological. However, an excess of glucocorticoids can negatively affect brain structures and generate disruptive effects in the offspring [14,15,88]. The effects of glucocorticoids on the fetus are less well known [98,99], although they play an important role in adults. The consequences of glucocorticoid exposure can vary greatly depending on the gestational age, severity and duration of exposure [99]. Effects in early and mid-gestation depend on the level of stress experienced by the mother. The hypothalamic–pituitary–adrenal (HPA) activity of the mother can also affect the glucocorticoid permeability of the placenta. Later in gestation, when the fetal HPA axis has developed functionally, fetal glucocorticoid concentrations can have effects independent of maternal levels [99]. Additionally, glucocorticoids act in a non-linear “U-shape function”, so the concentrations can have negative effects on emotionality and learning [100]. Glucocorticoids act directly on the development of the central nervous system, for example, in the hippocampus, a region of the brain characterized by high plasticity and having an important role in behavior and welfare [101].

One of the key protectors of the fetus during gestation is the placenta, once it can modulate the consequences of stressful events experienced by the mother and ultimately act as a buffer [102]. In the placenta, the 11β-hydroxysteroid dehydrogenase enzyme type 2 (11βHSD2) oxidizes the biologically active form of cortisol [102,103], helping the placenta to act as a protector barrier to deal with high levels of cortisol from the maternal organism exposed to a stressor. However, in the long-term, chronic stressful situations have the potential to inhibit the capacity to upregulate the type 2 enzyme activity. In other words, previous exposure to chronic stress affects the protective capacity of the placenta impacting 11βHSD2 activity [104].

The effects of prenatal stress may generate changes in offspring emotionality, since they impact negatively some brain structures such as the hippocampus and amygdala [96,105]. In order to evaluate the emotionality of non-human animals, tests have been validated [106]. These include the open-field test and the novel object test [14,106,107,108], in which behaviors such as activity, exploration and vocalization can be used as indicators of emotionality [109]. Interestingly, one specific type of stereotypy in sows expressed during the gestational period is related to decreased fear responses in their offspring [90,91], indicating that the sows expressing it were more adjusted and had a more organized environment for the development of their offspring. Moreover, the cortisol in non-stereotyper sows was higher compared with the sows performing stereotypy [90].

In rats (*Rattus norvegicus*), maternal enrichment increased exploratory behavior in both male and female offspring [68]. Positive prenatal experiences decrease global methylation levels in the hippocampus and frontal cortex, affecting the ontogenetic development in the offspring [68]. Sows maintained in a simple enriched environment during gestation (provision of straw in the final third) resulted in impacts in the offspring, such as modulation in the HPA axis and behavioral changes related to emotionality, indicating improved welfare [92]. The enrichment during gestation reduced aggressiveness, nosing behavior and the concentration of salivary cortisol in the offspring [92]. During fear tests, the difference in the behaviors was sex-specific: females whose mothers were maintained in an enriched environment explored more and showed less fear of a novel object compared with those kept in a conventional farming environment [92], while the males did not show a difference.

It could be that the changes in offspring behavior were not a direct effect of the prenatal environment, because the effects of that environment on the mother–infant relationship and early social context have been demonstrated [88,93,110,111]. Enrichment during gestation may change the mothers’ behavior or even alter traits such as anxiety, impacting the maternal behavioral repertory. The consequences in the offspring could then be related to the mothers’ behavior during lactation instead of an impact related to prenatal exposure.

## 5. Epigenetic Processes Driving Fetal Programming

With the increasing knowledge of epigenesis, it is possible to consider dynamic changes in DNA expression, since the individual and the environment form a constant and dynamic cause-and-effect relationship. What would be the consequences of an individual performing stereotypic behavior to cope with the environment for a long period of time? The interactions between motivational systems, including frustration, and emotions must significantly change the epigenome. Epigenetics can chemically modify chromatin and affect genomic transcription. Epigenetic modifications can be stable and cross generations, and they are also dynamic, changing in response to environmental signals and stimuli [112]. There are three main epigenetic mechanisms: DNA methylation (the covalent modification of cytosine with a methyl group), histone modification (acetylation, methylation, phosphorylation, ubiquitination) and microRNA (small non-coding RNAs that post-transcriptionally regulate gene expression) [112].

The epigenetic modification of chromatin is a key regulator of gene expression, growth and differentiation in all tissues, including the central nervous system [95,113,114,115,116,117,118]. DNA methylation at the sites of CpG dinucleotides, at selected genomic loci, might affect social cognition [119], learning and memory [120,121], as well as adrenal responses [112,122,123]. Nevertheless, transcriptional silencing is considered mainly to be an outcome of the hypermethylation of gene promoter regions [124]. Hypermethylation is associated with dysfunctional gene expression in a wide range of psychiatric disorders [122,125,126,127], such as autism spectrum disorder [97,118], schizophrenia [97], depression and Alzheimer’s disease [58,128,129].

From an evolutionary perspective, epigenetic changes give the organism a mechanism for instantaneous adjustment. In the brain, each modification requires epigenetic activity, which can be transient because of fluctuations in neurotransmitters, or persistent because of alterations in dendritic morphology or synaptic pruning [95]. Heritable epigenetic changes facilitate rapid adaptation to adversities but can also lead in a mismatch of physiological profiles to later-life challenges, thus enhancing disease risk [112]. The consequences of prenatal stress on brain structures such as the limbic system are changes in offspring emotionality [96,105]. There could also be effects on males during sperm generation that alter the offspring. It is now clear that epigenetic processes, such as DNA methylation, are mechanisms involved in normal and pathological brain function [118,123,130,131].

One of the mechanisms by which the phenotype has been changed, regarding stereotypy expression during gestation, is by alterations in the methylation patterns in the brain of the offspring. Environmental enrichment reduces stereotypies and changes the neuroepigenome in the offspring (Tatemoto, in press). Stereotypies expressed by the mother during the prenatal period changes methylation patterns in several genes that are involved in neuroplasticity and psychiatric disorders, specifically in the limbic system of the offspring. The pathways are involved in basic cell functions, increasing the effects on neurophysiological mechanisms and disease risk (Tatemoto, in press). There are already some data elucidating the transgenerational effect of environmental enrichment on gene expression in the brain of the offspring [68], so it would be interesting to know more about the effects of enrichment on stereotypy reduction during prenatal programming.

## 6. Conclusions

Stereotypies are widely used as animal welfare indicators, especially because the expression of the behavior can signal psychological states. However, there are questions about the longer-term consequences if animals express stereotypies: do the stereotypies help in coping? Can these individuals be better adjusted or is their later welfare worse than that of individuals who do not show the behavior? Some of the answers to those questions relate to studies showing that environmental enrichment reduces stereotypic behavior, providing for some needs and improving the welfare of the offspring. Going further in the timeline of stereotypy expression, during the prenatal period, stereotypies expressed by the mother can change the phenotype of the offspring, especially regarding emotionality. We propose that one of the mechanisms changing the phenotype of the offspring in response to enrichment and stereotypies is epigenetic changes in the limbic system of the brain. Responses, such as stereotypy, change over time. In the first stages of the response, stereotypies may be helping animals cope with challenges. However, it could have detrimental effects in a long-term. The neuroanatomical changes in individuals showing stereotypies could be an effect rather than a cause of the stereotypy. In this context, it would be interesting to design studies aiming to understand the stereotypies effects on the epigenome, across time. Additionally, it would be beneficial to combine stereotypy behaviors with other welfare indicators in a timeline to elucidate if the impact, as well as the capacity to help deal with challenges, changes depending on the time of expression.

## Figures and Tables

**Figure 1 animals-12-02504-f001:**
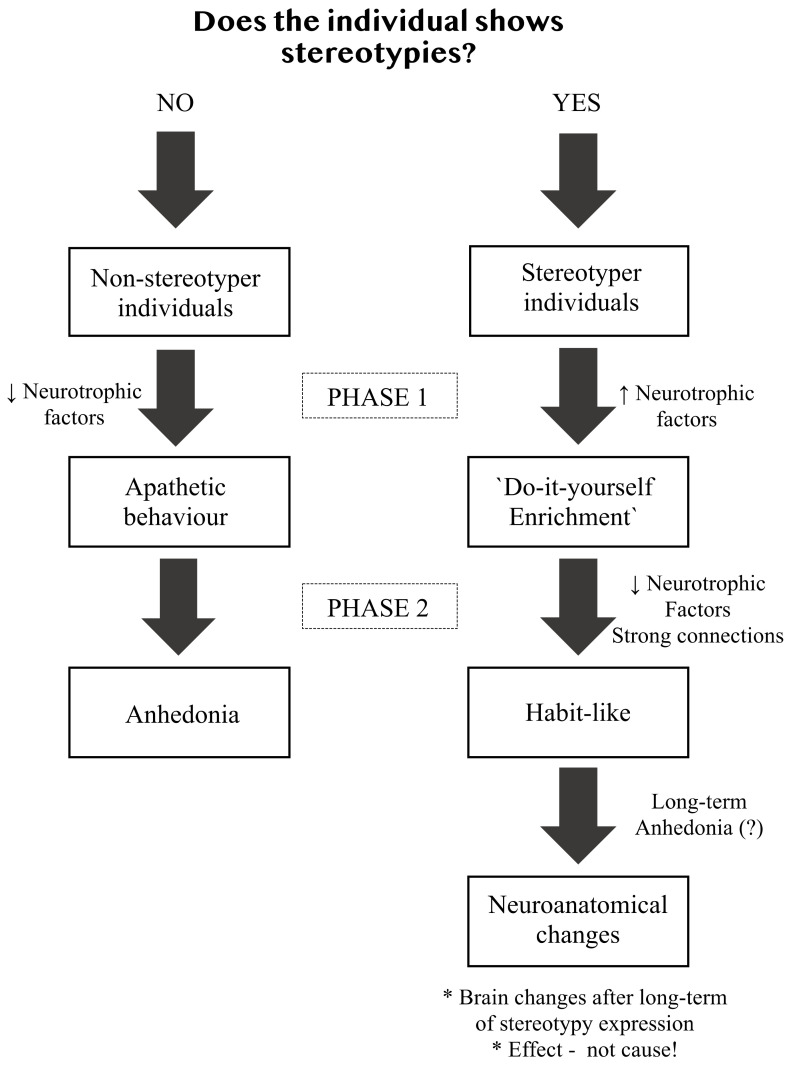
**The consequences of negative environments over time on the stereotypies expression.** Two possible pathways when the individual attempts to cope. Stereotypies could help to cope initially. In the long-term, it could be detrimental to reinforce the same connections as a habit, making a ‘scar’ in the brain. Also, in the long-term, it may have evolve into anhedonia. It is possible that neuroanatomical change types are a consequence, not a cause, of the stereotypy or apathetic behavior.

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
