# Peer review of "Changes in Stereotypies: Effects over Time and over Generations"

_animals, 2022, doi:10.3390/ani12192504_

Round 1

Reviewer 1 Report

I have no fundamental concerns about this paper. The main request is to clarify that this is a review supporting the hypothesis that stereotypies may be passed on to offspring via epigenetic mechanisms, and that this hypothesis needs to be tested further. The authors did a good job of using words such as “likely” and “could. However, explicitly clarifying in the Abstract, Introduction, and Summary that this is a proposed consequence of stereotypy will minimize the odds that it is subsequently cited as a fact that stereotypies are passed on to offspring.

Use caution with the word “stress”. I believe that in lines, 68, 258 and 102 “distress” may be the most appropriate word to use. Line 102, “stress response” is more appropriate. Line 218, clarifying the meaning of “stress resiliency” is recommended. Similar for line 253 and “prenatal stress”(stress response? Response to in utero stressors? Etc.). 

Author Response

Responses to reviewers

Reviewer1:

"I have no fundamental concerns about this paper. The main request is to
clarify that this is a review supporting the hypothesis that
stereotypies may be passed on to offspring via epigenetic mechanisms,
and that this hypothesis needs to be tested further. The authors did a
good job of using words such as “likely” and “could. However, explicitly
clarifying in the Abstract, Introduction, and Summary that this is a
proposed consequence of stereotypy will minimize the odds that it is
subsequently cited as a fact that stereotypies are passed on to offspring.

Authors: Thank you very much, all the suggestions were very much appreciated.

Exactly, and we mean that not necessarily the stereotypies pass to the next generation by epigenetic mechanisms (although there are genes involved as mentioned), but there are some effects in the emotionality showed in previous studies related to stereotypies expression of the mother during gestation, that affected the epigenetic changes in the limbic system in the offspring. So, the effects in the emotionality of the offspring could be related to epigenetic mechanisms, which we are in process to publish a manuscript elucidating and confirming this hypothesis – we cite in the topic 5, which the citation is “Tatemoto, in press”. It could be that the stereotypies as coping expressed by the mother during gestation, someway affects the epigenetic programming of the offspring.

Use caution with the word “stress”. I believe that in lines, 68, 258 and
102 “distress” may be the most appropriate word to use. Line 102,
“stress response” is more appropriate.

Authors: In line 68, we changed to distress. In line 102, we changed to stress response. Indeed, it is much more appropriated. Thank you!

However, in line 258, we are discussing a range of possibilities regarding prenatal stress experienced by the mother, which could not necessarily negatively impact the offspring. For this reason, we understand that would be more appropriate to keep stress based on the message we are trying to pass.  

Line 218, clarifying the meaning of “stress resiliency” is recommended. Similar for line 253 and “prenatal stress” (stress response? Response to in utero stressors? Etc.)"
Authors: We expanded the paragraph related to the suggestion of the line 218. Please, see below. 

“Male mice housed with enrichment after weaning gained less weight, displayed increased social behavior, presented lower corticosterone concentrations, and prefrontal IL-1β elevations in response to a mild social stressor. Additionally, they exhibited reduced TNF-α and increased brain-derived neurotrophic factor (BDNF) expression in the pre-frontal cortex [78]. In pigs, enrichment increased BDNF in blood [79] and BDNF has been linked with increased stress resilience [80], since enhance cognitive functions and is implicated in growth, maintenance, and plasticity in the brain [81]. This stress resilience is related to a buffer effect promoted by the environmental enrichment when animals are challenged, which agrees with other studies [75,82,83]”.

Regarding the second suggestion, related to the sentence at line 253, we expanded the explanation in the paragraph that precedes it, because we found more appropriated since we start to mention prenatal stress. In this sentence, we expanded the explanation about the prenatal stress.

“Prenatal environments are critical for fetal development, including the organization of the central nervous system. When mothers are stressed there can be a wide range of effects on offspring physiology and behavior, since the systemic physiological stress response of the mother has connection to the offspring that is in development in the uterus.”

Reviewer 2 Report

Dear Authors,

Thank you for sending this paper on stereotypy development to review. This is an interesting and engaging paper that summarises many of the key texts regarding stereotypy. The points relating to maternal influences on stereotypy were especially interesting. There are several points which I have attached relating to word choice, scientific name, and referencing errors. Otherwise, this is a well formatted and useful manuscript.

Author Response

Reviewer2:

"Dear Authors, thank you for sending this paper on stereotypy development to review.
This is an interesting and engaging paper that summarizes many of the
key texts regarding stereotypy. The points relating to maternal
influences on stereotypy were especially interesting. There are several
points which I have attached relating to word choice, scientific name,
and referencing errors. Otherwise, this is a well formatted and useful
manuscript."

Authors: Thank you, it was very much appreciated. We have addressed all the suggestions, and other minor changes to fit better in the new version of the manuscript. We are sending the pdf version with the answer in each comment.
